# Augmented Reality-Assisted Deep Reinforcement Learning-Based Model towards Industrial Training and Maintenance for NanoDrop Spectrophotometer

**DOI:** 10.3390/s23136024

**Published:** 2023-06-29

**Authors:** Hibah Alatawi, Nouf Albalawi, Ghadah Shahata, Khulud Aljohani, A’aeshah Alhakamy, Mihran Tuceryan

**Affiliations:** 1Department of Computer Science, Faculty of Computers and Information Technology, University of Tabuk, Tabuk 47512, Saudi Arabia; 431007209@stu.ut.edu.sa (H.A.); 431000011@stu.ut.edu.sa (N.A.); 431010355@stu.ut.edu.sa (G.S.); 431010491@stu.ut.edu.sa (K.A.); 2Artificial Intelligence and Sensing Technologies (AIST) Research Center, University of Tabuk, Tabuk 47512, Saudi Arabia; 3Department of Computer Science, School of Science, Indiana University-Purdue University, Indianapolis, IN 46202, USA; tuceryan@iu.edu

**Keywords:** augmented reality, extended reality, reinforcement learning, training, maintenance, localization, spectrophotometer

## Abstract

The use of augmented reality (AR) technology is growing in the maintenance industry because it can improve efficiency and reduce costs by providing real-time guidance and instruction to workers during repairs and maintenance tasks. AR can also assist with equipment training and visualization, allowing users to explore the equipment’s internal structure and size. The adoption of AR in maintenance is expected to increase as hardware options expand and development costs decrease. To implement AR for job aids in mobile applications, 3D spatial information and equipment details must be addressed, and calibrated using image-based or object-based tracking, which is essential for integrating 3D models with physical components. The present paper suggests a system using AR-assisted deep reinforcement learning (RL)-based model for NanoDrop Spectrophotometer training and maintenance purposes that can be used for rapid repair procedures in the Industry 4.0 (I4.0) setting. The system uses a camera to detect the target asset via feature matching, tracking techniques, and 3D modeling. Once the detection is completed, AR technologies generate clear and easily understandable instructions for the maintenance operator’s device. According to the research findings, the model’s target technique resulted in a mean reward of 1.000 and a standard deviation of 0.000. This means that all the rewards that were obtained in the given task or environment were exactly the same. The fact that the reward standard deviation is 0.000 shows that there is no variability in the outcomes.

## 1. Introduction

Augmented reality (AR) technology enhances the impression of real-world objects by overlaying information over the user’s perception. An AR system generally projects computer-generated augmentations over actual items, merging real and virtual elements [1]. For industrial applications, AR is an established technology [2,3]. Equipment maintenance is a set of procedures that help keep equipment in good operating order. Maintenance may also help manage a variety of other difficulties. A contemporary industrial firm often employs three maintenance methods: by schedule, by operational time, and by actual state.

During maintenance, the equipment parameters such as hourly rate, amount of processed goods, mileage, and so on are taken into consideration based on the operational time. During maintenance, several parameters are monitored based on their current status [4,5]. As a result, being able to foresee machine maintenance activities and complete them in a timely manner may lead to effective troubleshooting while also increasing machine tool availability. Furthermore, because the replacement of the failure mode may account for up to 70% of overall maintenance costs, based on that premise, significant research effort has been made to address the development and design of real-time maintenance tool kits and mobile device-based applications to avoid needless errors from occurring [6]. As a result, in the I4.0 age, maintenance personnel should be assisted by linked instruments and machine intelligence to effectively accomplish their activities without the need to know the precise position of machines or detailed knowledge about the programmable topology. Typically, maintenance operators are highly skilled and undertake maintenance activities by following paper-based technical drawings [7].

There are several studies and reviews that discuss various aspects of AR technology to develop a user-friendly platform. It also explores the challenges that are related to its adaptability. They explore the potential applications of AR technology in the industrial sector which include improving the virtual interface, identifying and training workers, and facilitating collaboration and maintenance among streams in an industrial context [8,9,10,11].

In this research, we present an efficient industrial AR-assisted deep RL-based model [12] that can find a component in a system and view maintenance instructions for relevant failure modes on the operator’s smartphone in a portable and real-time approach. Smartphone sensors provide user position with a certain degree of accuracy, but this accuracy does not seem to be enough to prevent errors in the localization and tracking of small objects found in devices such as the NanoDrop Spectrophotometer. To solve this issue, we suggest the use of RL. The agent’s ultimate objective is to learn a policy that maximizes the expected cumulative reward over time, which involves making decisions that lead to the highest possible rewards in the long run. The selected maintenance scenario in our case study focuses on DNA sample localization and how to fix the position to the correct angle and alert the user in case of failure. This paper’s main contributions can be summarized as follows:An assistance system is available for untrained operators to perform maintenance tasks.The platform can be accessed on smartphones and tablets, eliminating the need for expensive AR hardware.The system can locate assets within the complex industrial shop floor without human intervention.By replacing paper-based instructions with an AR-assisted deep RL-based model, a digital feature to reduce retrieval times.Our technology is anticipated to enhance communication between manufacturers and maintenance personnel in the future.

This study aimed to use the advantages of AR and RL technologies over traditional methods to show the need for similar research. A system that combines training and maintenance using an AR-assisted deep RL-based model which can provide all the industrial services with the ability to (1) increase user engagement and learning, (2) higher knowledge retention, (3) reduce training and repair costs, (4) overcome distance, and (5) create a safe environment of dangerous situations for novice learners.

## 2. Literature Review

In this section, we discuss the literature related to AR in training and maintenance, the use of RL with XR, manufacturing data, and strategies for visualization enhancement in Industrial AR.

### 2.1. AR Technologies in Education and Training

Nowadays, these technologies are already gaining ground in a number of industries [3]. Through the use of XR offerings, businesses, and organizations can improve their worker training and develop their trainees’ understanding of the content. However, it is not enough to replace the need for in-person learning but they can be used hand in hand to advance learning and training. Due to the increasing popularity of immersive learning, many educators are now using XR to provide their learners with the necessary experiences. Aside from being beneficial, it can also help trainees retain their knowledge. For instance, to improve the training of first aid students for infants with airway obstruction, the study by Chen and Liou [13] used AR to create an assessment system that allows one to perform a quantitative assessment in a digital setting. The system can be used in combination with other training methods to help instructors reduce the time and cost of training as well as help students improve their skills in emergencies.

To build a systematic approach to increase the functionality and users’ acceptance of future XR systems, a framework for designing XR systems inside a manufacturing setting is presented to show that the manufacturing sector may advance past the “wow impact” of XR demonstrators and reach the point where XR systems could be fully implemented and enhance the ways in which people go about their daily business. For XR systems to be more widely used in manufacturing, it is crucial to guarantee their usability and user approval; not only can XR systems deliver the claimed advantages but also entirely new methods of interacting with computers for both end users and system developers [14]. One of the most important and promising improvements in the architecture, engineering, and construction (AEC) sectors is the use of building data modeling (BIM) as a digital information management system so that design data may be digitally managed for use across a project’s life cycle [15].

### 2.2. AR Technologies in Maintenance and Repair

Apart from providing training, AR can also be utilized by organizations to maintain and service their equipment. Through 3D models, individuals can interact with industrial tools by increasing their sizes, examining their internal structures, and rotating them. AR tools can also provide guidance and instruction in real time. Despite the hype surrounding the Fourth Industrial revolution, some people still believe that such innovations are not practical, but there are reasons to embrace them. In the maintenance and repair sector, AR could support the traditional method of work and ease the process. Various factors such as the rising number of AR hardware, as well as the lower cost of development, are expected to drive up innovations in this area [16]. For instance, a cloud-based database was developed to collect information about equipment maintenance and inspection. It was then integrated with AR technology to allow users to perform inspections and maintenance on their mobile devices [17].

The proposed system by Malta et al. [18] organizes and executes service requests for automobile engines, which are a set of instructions for the maintenance specialists to follow. A technician wears immersive VR glasses while using a system that is based on YOLOv5, a deep neural network that recognizes mechanical parts in car engines and provides directions according to the work order. Users look through the glasses to see the car parts and the updated real-time directions for how to proceed. Furthermore, Idriss et al. [19] explored the use of AR in a pilot project to help novice learners improve their knowledge of predictive maintenance. This case study is designed to assist a student in an industrial asset management while learning about machine fault simulators and the analysis of vibration data. Through the use of AR technology, users can inspect an aircraft component using a 3D model. They can then see the synchronized data from the digital twin database. Inspections and maintenance can be performed using this mixed reality vision. The data related to a particular part can easily be attached to it. For instance, the locations of damage can be marked by the inspector [20].

### 2.3. The Use of Reinforcement Learning with Extended Reality

The field of RL is booming in AI which can be used in fields such as finance, gaming, and medicine [21,22]. The optimization of task and resource allocation for each user is a complex problem that needs to be solved to minimize energy consumption. Xing et al. [23] presented a deep learning algorithm based on a multi-agent deep deterministic policy gradient (MADDPG) that can be used to enable a multi-user wireless network by offloading the computation task to the mobile edge computing (MEC) server, reducing the latency and energy consumption of the user terminal for AR application. The study designed by Chengxi et al. [12] provided a framework for safe symbiotic human–robot interaction (HRI) by integrating various features such as visual augmentation, velocity control, and collision detection. The proposed method utilizes deep RL for collision avoidance and is enabled through AR. In addition to providing a framework for safe symbiotic HRI, Hetzel et al. [24] demonstrated the potential of RL to improve the performance of mid-air typing. Many studies have been conducted on enhancing the service experience by providing ideal feedback. The cyber-physical system (CPS) proposed by Choi et al. [25] can be used with various sensors to create an interactive environment.

The goal of the Kang and Heo [26] in their study was to develop an automated and efficient method for the quick and accurate 3D eye center tracking and detection. It utilizes a single camera and near-infrared (NIR) light-emitting diodes. The proposed method includes eye-nose detection, shape keypoint alignment, tracker checker, and the LED on/off control. It can generate facial subregion boxes by analyzing the nose and eyes using an error-based learning algorithm. Chan et al. [27] were able to improve the image quality and computational efficiency of 3D generative adversarial networks (GANs) capable of handling various features by decoupling the generation of 2D convolutional neural networks (CNNs) and neural rendering. The framework was also able to inherit the efficiency of these generators and perform various experiments with 3D aware models.

Despite the applications of deep RL, it still cannot yet be feasibly fully implemented in intelligent vehicle control systems. However, cities are becoming more intelligent with the help of AI-based systems [28]. To improve road safety, a deep RL agent is being developed through a collection of machine learning (ML) concepts. The results of this study indicate that the ML models can effectively replace the fuzzy classifier [29]. The Pina et al. [22] study aimed to investigate the effects of RL on the path planning tasks performed in intelligent vehicle control systems. We then present two potential solutions to address the gap between the current theoretical and practical applications of deep RL.

### 2.4. Manufacturing Data and Strategies for Visualization Enhancement in Industrial AR

In-depth descriptions of individual AR applications built to verify their viability in a range of industries are provided in several articles [2]. They present and evaluate the viability of using mixed reality (MR) to enhance the communication of safety issues on worksites through a holographic program that provides a collaborative setting through a head-mounted display (HMD) [30] in which others can view and interact [31]. Two tests provide an empirical comparison of 11 bare-handed, mid-air mode-switching approaches appropriate for virtual reality (VR); the first analyzes seven hand movements, both dominant and nondominant, and the second uses four methods to investigate the impact of using a dominant hand device for selection as well as other nuanced dominant-hand strategies [32].

The voxelization approaches offer additional functionality for both the display of 3D objects as well as for the simulations of complicated geometries, particularly in the context of nontraditional manufacturing processes, as was evident from the papers that were under review [31]. The Microsoft HoloLens MR with the robot operating system (ROS) used by Mourtzis et al. [33] operates a robotic arm remotely in almost real time while also presenting safety areas on the shop floor considering the technician’s range of view. Health and safety were cited as the top advantages of remote user research by Ratcliffe J. et al. [34], who also mentioned more general concerns including air quality in enclosed lab areas and the possibility of COVID-19 transmission through shared HMDs and controllers. Recent approaches for curve design [35] and attribute transmission among similar triangle meshes [36] have also been used to offset volumes or shells around triangle meshes. Table 1 summarizes the outcomes of some studies representing the I4.0 type, software, hardware, limitations, and challenges. These studies were selected due to their inspiration and impact on this research methodology and the use of similar I4.0 techniques, software, hardware, and for model evaluation with other studies in Section 5.2.

## 3. Methodology

This section describes the prototype of the proposed system based on AR-assisted deep RL-based model. The process starts with image scanning for feature extraction and matching, then an explanation of how ML agents, 3D modeling, AR SDK, 3D engine, tracking camera, and smartphone sensors can be utilized to create the interactive system.

### 3.1. Framework Conceptual

As can be seen in Figure 1, the application starts with (1) scanning an image or object model; to (2) obtain the required frame; for (3) feature extraction; which then (4) is matched with (5) scaling and orientation based on the (6) operator action to represent a (7) 3D visualization on the screen. To enhance the object localization and tracking in our system, the (8) RL algorithm was used in the environment.

Initially, the developers created a target image, which represents an object to be recognized in the system. This image is then uploaded to the cloud recognition server. When the user runs the app, the device’s camera is activated, and the SDK starts looking for the target image. As the camera captures the video frames, the SDK analyzes each frame to detect the target image. It uses feature detection algorithms to identify unique points in the target image that are also present in the camera frame. The SDK then uses these points to estimate the position and orientation of the target image relative to the camera. This process is called pose estimation. Once the SDK has estimated the pose of the target image, it can overlay AR content on top of the real-world image in the camera view. The framework concept is based on two basic modules: first, training on how to use a certain device, and second, for device maintenance and repair. At the beginning of this process, the user can choose which module is needed to proceed and explore the next steps.

### 3.2. Training and Maintenance Setups

The use of XR technology can assist organizations not only with training but also with keeping their equipment in good working condition. Three-dimensional models enable users to explore the internal structure and size of the equipment. Although some people still view these technologies as impractical, they are essential for organizations seeking to improve their operations. In the repair and maintenance industry, AR can boost efficiency by enabling more effective repairs. As the number of AR hardware options increases and development costs decrease, the adoption of this technology is expected to rise. Companies that recognize its value should incorporate it into their operations.

For the current system, we created two scenario setups: (1) the training setup, which provides the user with the main guidelines on how to use the device without the need for an instructor; and (2) maintenance setup, where we assume a case of failure and present the step-by-step instructions solution without an expert. However, the repair scenario could include an option for contacting an expert and sharing a live feed of the problem. Both scenarios for our case study are explained in Section 4.

To generate manufacturing information suitable for job aids in mobile AR, it is necessary to collect 3D spatial information and the class and location of each piece of equipment. The 3D models must be matched exactly to the physical parts to provide a seamless AR experience. Calibration, achieved through the binary marker tracking technique, is crucial for integrating the two components. Once calibrated, all 3D models in the virtual parts can be superimposed on their corresponding physical counterparts.

This module is centered on a precedence automated system that assists engineers in creating interactive AR guidelines for disassembly, assembly, and inspection tests. The guidance steps are based on a knowledge base stored and updated on a cloud server. The recommended technique can recognize and segment services and anticipate their 3D depths and geographic data, even when the real equipment is dynamically capable of functioning or moving by altering its configuration. This updates the location of the equipment, ensuring that the product information is correctly positioned around the target.

### 3.3. Feature Extraction and Matching

Feature extraction and matching in the 3D engine is a powerful technique that allows developers to create accurate and engaging XR experiences. By accurately tracking unique features in the game environment, feature extraction and matching can create a more immersive and engaging experience for agents, but it also requires the careful consideration of the challenges associated with the technique. Feature extraction and matching is a technique used to identify and track specific features in an environment. This technique is commonly used in XR applications and is essential for creating accurate and engaging XR experiences.

Feature extraction and matching involve identifying unique environmental features, such as corners, edges, or other distinctive patterns. These features are then tracked in real time as the agent moves around the environment, allowing the target to accurately track the agent’s movements and adjust the XR experience accordingly. One common approach to feature matching and tracking in Unity is to use computer vision techniques such as scale-invariant feature transform (SIFT) [37] or speeded up robust features (SURFs) [38]. These techniques involve identifying unique features in the game environment and creating a feature descriptor that can be used to match the feature across multiple frames.

### 3.4. Orientation and Scale

To project the associated CAD model onto the picture plane of a mobile device, the real posture of the 3D model must be calculated, including its orientation and scale. The proposed method orients the detected features by comparing and contrasting the gradients of the target model’s corresponding features. The distance between the neighboring ones also affects the scale calculation. This method produces a rotation matrix and a translation vector for displaying the 3D model’s points on a mobile device.
(1)$$xc=\left[R|t\right]xX\to \left[\begin{array}{c} x\\ y\\ z \end{array}\right]=\left[\begin{array}{cccc} r1 & r2 & r3 & tx\\ r4 & r5 & r6 & ty\\ r7 & r8 & r9 & tz \end{array}\right]\left[\begin{array}{c} X\\ Y\\ Z\\ 1 \end{array}\right]$$

The projection of a point in image coordinates (xc) is determined by its real-world coordinates (X) and the pose matrix [R|t]. The intrinsic matrix of the camera is used to align the pixel coordinates of the camera with the coordinate projection:(2)$$K=\left[\begin{array}{cccc} f & Y & px & 0\\ 0 & f & py & 0\\ 0 & 0 & 1 & 0 \end{array}\right]$$

The transformation matrix is computed by taking into account the focal length (*f*) of the axes *x* and *y* and the zero-skew factor between them. The transformation matrix is then obtained after taking into account the (*K*) value of the calibration matrix.
(3)$$M=K\times T\to M=\left[\begin{array}{cccc} f & Y & px & 0\\ 0 & f & py & 0\\ 0 & 0 & 1 & 0 \end{array}\right]\times \left[\begin{array}{cccc} r1 & r2 & r3 & tx\\ r4 & r5 & r6 & ty\\ r7 & r8 & r9 & tz \end{array}\right]$$

The posture matrix is represented by *T*. Ultimately, the camera image plane equivalent coordinates (Xf) for a point with real-world coordinates Xi are determined:(4)Xf=M×Xi

### 3.5. Reinforcement Learning Algorithm to Enhance Localization and Tracking

The reason for using the RL algorithm is that it is not possible to track more than one object, so this object is the NanoDrop Spectrophotometer and the agent is the pipette device. The presented model is a well-designed and fine-tuned implementation of the proximal policy optimization (PPO) algorithm with a set of carefully chosen hyperparameters that enable dynamic control of the training process. PPO is a widely used approach for training RL agents due to its stability, robustness, and scalability. The model incorporates additional features such as normalization, layer depth, and encoding type, which can effectively enhance the performance of the agent. The hyperparameters, including batch size, buffer size, learning rate, and the number of epochs, are selected based on prior experimentation and analysis. Moreover, this model employs a learning rate schedule, beta schedule, and epsilon schedule, which allows for the dynamic adjustment of these hyperparameters over time, catering to the requirements of complex and dynamic environments.

The reward signals and goal conditioning are also important components of the model, allowing the agent to learn from feedback and adjust its behavior accordingly. The extrinsic rewards with a gamma of 0.99 and a strength of 1.0 are commonly used for RL tasks, while the incorporation of goal conditioning can effectively improve the learning efficiency and effectiveness of the agent.

The model that we presented in this study is a customized implementation of PPO, which allows for dynamic training control. The incorporation of additional features and reward signals, as well as goal conditioning, allows for effective learning and adaptation of the agent. This research provides valuable insights into the design and optimization of RL models and has implications for future research and development in this area.

#### Proximal Policy Optimization

The PPO algorithm used in ML agents is an on-policy algorithm that updates the policy parameters using a clipped surrogate objective function. The algorithm performs the multiple iterations of collecting data from the environment, computing the advantage estimates, and optimizing the policy using stochastic gradient descent. The PPO used in ML agents is a variant of the original PPO algorithm proposed by Schulman et al. [39]. The clipped surrogate objective function is based on the ratio of the new policy to the old policy, and it is clipped to ensure that the policy update is manageable. This helps to prevent the policy from diverging during training and stabilizes the learning process. Moreover, ML-Agents PPO also uses a value function to estimate the expected return, which is used to compute the advantage estimates. The value function is learned using a mean-squared error loss function, and it is simultaneously updated with the policy parameters.

To stabilize the learning process in PPO, the clipped surrogate objective function is an effective modification of the original objective function. The clipped surrogate objective function is used to update the policy parameters in PPO. The objective is to maximize the expected total reward of the agent over a sequence of actions. The clipped surrogate objective function is a modification of the original objective function that helps prevent the policy from diverging during training and stabilizes the learning process.

The clipped surrogate objective function is based on the ratio of the new policy to the old policy. The ratio measures how much the new policy has changed compared to the old policy. The objective is to maximize the expected total reward subject to a constraint on the maximum change in policy. The clipped surrogate objective function is defined as follows:(5)$$L^{clip}\left(\theta \right)=min\left(\frac{\pi \theta (a|s)}{\pi \theta_{old}\left(a|s\right)},clip\left(1-\epsilon ,1+\epsilon ,\frac{\pi \theta (a|s)}{\pi \theta_{old}\left(a|s\right)}\right)\right)A\left(\pi \theta_{old}\right)$$
where θ is the new policy parameters, πθ(a|s) is the probability of taking action a in state s under the new policy, $\pi \theta_{old}\left(a|s\right)$ is the probability of taking action a in state s under the old policy, $A(\pi \theta_{old})$ is the advantage estimate of the old policy, and ϵ is a small constant that controls the degree of clipping.

The clipping operation restricts the ratio to lie within a certain range, [1−ϵ,1+ϵ]. This ensures that the policy update is not too large and prevents the policy from diverging. If the ratio is within the clipping range, the clipped surrogate objective function becomes the ratio multiplied by the advantage estimate. If the ratio is outside the clipping range, the clipped surrogate objective function becomes the clipped value multiplied by the advantage estimate.

For more clarification, the variables in this equation are defined as follows: $L^{clip}\left(\theta \right)$: the clipped surrogate objective function used in PPO to update the parameters of the policy network. θ: the parameters of the current policy network that are being updated during training. πθ(a|s): the probability of taking action *a* in state *s* according to the current policy network with parameters θ. $\pi \theta_{old}\left(a|s\right)$: the probability of taking action *a* in state *s* according to the previous policy network with parameters $\theta_{old}$. clip(1−ϵ,1+ϵ,x): a function that clips the value of *x* between 1−ϵ and 1+ϵ. $A(\pi \theta_{old})$: the advantage function, which measures how much better or worse a particular action is compared to the average action taken in *a* given state under the previous policy network.

Thus, the equation does not explicitly refer to the agent; rather, it is used to update the policy network that the agent is using to interact with the environment.

### 3.6. Visualization of 3D Models

In the maintenance and training operations, 3D objects are employed to display the relevant stages of the maintenance task in an easily understandable and concise manner. It is important to establish the appropriate resolution and terms of use for 3D objects when creating them. Usually, high-quality designs are not required. Additionally, using low-quality designs ensures that the object remains compact, enabling the incorporation of more 3D objects in our methodology. The file size of an object is also affected by the number of materials, textures, and polygons that it contains. The primary consideration for creating 3D objects is the proposed method, which is capable of operating smoothly on most mobile devices with decent processing power.

Hardware description. The primary computing device was an HP Pavilion laptop with a 7th generation SSD Core i7 processor and a 2 terabyte hard disk, providing ample storage and processing capabilities. We also used a Huawei Nova Y70 smartphone with 128GB ROM and 4GB of RAM, model MGA-LX9, to test the mobile compatibility of our software application. This device provided a high-quality user experience with a large display and sufficient storage and processing power to support our software. We used a camera HP 320 wired USB with a 1080 FHD black for testing the model target.

Software description. The system was built using the Unity 3D engine, a highly versatile and popular game development platform known for its robust features and cross-platform compatibility. To implement RL algorithms, we utilized the ML-Agents open source toolkit available on GitHub. For training the model, we employed the Anaconda software, a widely used package manager, and an environment manager in Python. The software was programmed using a combination of C# and Python, two powerful programming languages with extensive libraries and toolsets for developing complex applications. To enhance the realism and interactivity of the AR environment, we incorporated ARCore and Vuforia, two popular AR development platforms. These tools allowed us to create immersive AR experiences by overlaying 3D models in the real world. Additionally, we used Photoshop to refine and enhance the 3D models used in the application.

## 4. Experimental Results

We generate image-based and object-based data augmentation for our case study: NanoDrop Spectrophotometer. In this section, we present the results of our training and maintenance scenario setups.

### 4.1. Use Case: NanoDrop Spectrophotometer

The NanoDrop Spectrophotometer was chosen for laboratory significance. The proposed system is designed to lead the user (technician) through each process, displaying information and instructions in the AR application. The 3D model of the Nanodrop Spectrophotometer is shown in Figure 2. The Thermo Scientific NanoDrop Spectrophotometer device allows one to carry out more analyses in less time with the same amount of resources without compromising the reliability of the single-sample model. The NanoDrop Spectrophotometer can take full-spectrum UV–Vis measurements of up to eight different DNA samples at the same time. With an eight-channel pipette, one can easily measure up to 96 different samples in six minutes using a linear array of pedestals [40,41].

The NanoDrop line uses a unique technology that lets a sample be pipetted onto an optical surface. This method results in the presence of surface tension, which maintains the volume of the sample while the measurement is being performed. Following the measurement, the surfaces are thoroughly cleaned with a lint-proof lab wipe. It features an innovative software platform that allows users to create custom reports and export data with ease. It has a built-in sample position illuminator that helps reduce errors and improve the efficiency of ones’ operations in environments such as biorepositories and laboratories that focus on quality control [42,43].

The experimental procedure includes two main scenarios: (1) the training scenario aims to provide a comprehensive guide on how to use the NanoDrop Spectrophotometer device without requiring an instructor; and (2) a maintenance scenario, typically assuming a failure and providing a step-by-step procedure for addressing the issue without an experienced technician. In a repair scenario, the user can potentially contact an authorized specialist and share a live feed of the problem.

### 4.2. Training Scenario Results

In XR training, AR technologies can provide real-time guidance and support to workers by performing basic and regular tasks. A worker can use an AR application to visualize the digital overlays of training instructions and procedures superimposed onto the physical equipment they are working on, as shown in Figure 3 and Figure 4b. This can help reduce the time and cost associated with training tasks as well as improve the accuracy and efficiency of training processes. The NanoDrop Spectrophotometer guide is designed to help novice learners understand the following aspects: (1) device parts and their functionalities; (2) electrical outlets; (3) sensors and sampling ports.

Additionally, the guide provides step-by-step instructions on how to add samples, handle the Pipette tool and tips, create a 3D design for each part, and add DNA samples. The training plan provides an AR guide for how to use the device. There are two technologies, optical see-through (OST) and video see-through (VST) [44]. The basis of VST is to use the camera to frame the surrounding environment. The following adds virtual templates or icons to the real-time streaming video. OSTs work in a different way, showing virtual models mounted on partially transparent lenses. In this scenario, the user sees the original external environment with a default form added to the environment.

However, when using VST technology, the viewer sees the actual world on the screen of the device so that a set of indicators appears on each part of the NanoDrop Spectrophotometer showing the function and importance of each part when clicked. Figure 4a shows the primary training interface consisting of a two-button layout and a progress bar: (Figure 3a) shows the text instruction of one of the steps on how to use the NanoDrop, and (Figure 3b) contains a green marker on the physical parts of the device. When you click on it, photo instructions about the way to use each part appear step by step; (Figure 3c) represents an animation instruction on the use of a Pipette. The system proposes a comprehensive educational guide that will replace the traditional handbook manual.

### 4.3. Maintenance Scenario Results

A realistic use case scenario involving a NanoDrop Spectrophotometer, supplied by a collaborating manufacturer was used to evaluate the effectiveness of our proposed methodology. The current maintenance procedure features detailed, step-by-step instructions for completing the entire process. Then, the operators were given a real-time demonstration of the maintenance process using the application. After conducting extensive research on the NanoDrop Spectrophotometer, several common maintenance issues were identified and applied in our system, including the following: Issue 1: verifying that the wires in the port are properly connected; Issue 2: adjusting the angle of the pipette when adding samples; and Issue 3: ensuring that the device’s top cover is closed to be ready for the next step, as can be seen in Figure 5.

To assist users in resolving these issues in an interactive manner, the system provides guidance and support. The camera records activities within the industrial site and serves as an input sense device for the planned AR-assisted deep RL-based model. The origin of interest in an object is first discovered at the level of the picture. This process is carried out by a tracking algorithm that follows the dimensions and position of the object for every recording frame. The suggested system will give the operator the necessary guidance as they start the maintenance procedures on the equipment. The system can be easily manipulated by the user through the buttons located on the asset. After the maintenance is complete, the system will send a confirmation message.

Overall, the process of setting up error detection involves creating a target image, configuring the SDK’s settings, and handling errors, whilst RL can be used for estimating the position of an agent in a dynamic environment. This can be achieved by training the agent to make decisions based on feedback from the environment, in the form of rewards or punishments, and using those decisions to update its position estimate. The value represents the immediate reward that the agent receives for its actions. The ultimate objective of the agent is to learn a policy that maximizes the expected cumulative reward over time, which involves making decisions that lead to the highest possible rewards in the long run.

### 4.4. System Model with Reward Estimation through Active Learning: Step-by-Step Results

Agents are able to acquire and retain knowledge through the use of deep RL. Training the ML agent using the PPO algorithm [39] by considering the agent is a pipette device and the target is a NanoDrop Spectrophotometer shown in Figure 5, and the goal is to collect the target with a reward of +1 and a fail of −1 with four actions identified. The start is defined as the episode at the beginning of the iteration when the target and agent are in a particular position. After the first iteration episode resets, it starts collecting observations to identify the target position to determine the suitable action of the agent to collect rewards. Multi-agents were created to enhance the training and maintenance processes.

The goal of the proposed model is to maximize the communication reward that the agent receives from the mobile AR edge computing system. To achieve this, the system is modeled as a decision requester process that aims to identify the optimal solution. This process comprises a four-tuple Markov decision model (S,A,R,P). S{s1,...,sN} is state space and si=(Ii,xi). Whereas A={a1,a2,...,ai} refers to the action space, ai is the channel gain, and *I* is the number of channels. R:S×A is the reward function. *P* is the transition probability. In addition, let π=P(a|s) be the optimal position that should be determined by policy and all agents. These agents should then use the distance between their target and local positions to collect rewards. Figure 6 depicts an animated movement of the pipette device as an RL agent on top of the NanoDrop Spectrophotometer as a target that keeps moving until the pipette reaches the correct position. These animated instructions help novice learners maintain and adjust the pipette angle while adding the DNA samples.

Table 2 indicates that the performance of the system improves as the number of steps increases. The mean reward starts with negative values, indicating low performance; however, it gradually increases and reaches the maximum value of 1.000 at the final step. Meanwhile, the standard deviation of rewards decreases over time, suggesting that the system becomes more consistent in achieving positive outcomes. The increase in time elapsed for each step is expected, as the system requires more time to complete each trial or episode. The standard deviation of rewards provides further insight into the distribution of rewards. A high standard deviation suggests a wide distribution of outcomes, with some trials resulting in significantly worse rewards than others. In the present case, the standard deviation is approximately half the magnitude of the mean reward, indicating some concentration of negative rewards around the mean; however, there is still a notable level of variability in the outcomes. Overall, the summarized performance metrics in Table 2 offer valuable information for monitoring progress and identifying opportunities for system enhancement.

If the mean reward is 1.000 and the standard deviation of the reward is 0.000, this means that all the rewards obtained in the given environment or task are exactly equal to 1.000. The fact that the standard deviation is 0.000 indicates that there is no variability in the rewards obtained. In other words, every trial or episode in the task results in the same reward of 1.000. This situation is relatively rare in practice, as most tasks and environments involve some degree of variability in the rewards obtained. However, if this were the case, it would suggest that the agent (Pipette) is performing very well and achieving the maximum possible reward in every trial.

The mean reward is calculated as the average of the rewards. Mathematically, it can be represented as:(6)$$Q_{meanReward}=\frac{1}{n}\sum_{i=1}^{n}R_{i}$$
where *n* is the number of episodes, $R_{i}$ is the reward obtained in the episode, and $\sum_{i=1}^{n}$ denotes the sum of rewards over all trials of the agent (pipette).

The standard deviation of rewards is a measure of the variability or spread of the rewards obtained. It is calculated using the following equation:(7)$$StdOfRewards=\sqrt{\frac{1}{n}\sum_{i=1}^{n}{\left(R_{i}-Q_{meanReward}\right)}^{2}}$$
where *n* is the number of trials of the pipette (agent), $R_{i}$ is the reward obtained in the ith trial, $Q_{meanReward}$ is the mean reward calculated using Equation (Equation 6), and $\sum_{i=1}^{n}$ denotes the sum of squared differences between individual rewards and the mean over all episodes. This formula computes the square root of the average squared differences between the rewards and their mean. The result is a measure of how far the rewards are on average from the mean value, indicating the level of variability in the rewards obtained in the model.

## 5. Evaluation Discussion

Upon the completion of detection, the AR technologies produce lucid and easily comprehensible instructions for the maintenance operator’s device. The system’s design adheres to virtualization, modularity, and service-oriented principles, resulting in a customized digital maintenance service that can be adapted to suit diverse maintenance scenarios in manufacturing. The application’s efficiency was assessed on a real NanoDrop Spectrophotometer in the industry, establishing that the incorporation of 3D visualization for maintenance instructions to enhance the operators’ user experience during maintenance tasks.

### 5.1. Position of Virtual versus Real Objects

During the training process of an RL model, it is crucial to balance exploration and exploitation. Exploration refers to the process of trying out different actions to discover new information about the environment, whereas exploitation refers to the process of choosing the most effective actions based on the current knowledge of the environment. Maximizing the policy entropy is a common technique used to encourage exploration during the early stages of training. By maximizing entropy, the policy is encouraged to explore more possibilities and avoid getting stuck in a suboptimal solution. As the policy becomes more familiar with the environment and accumulates more experience, it should shift towards exploiting the most effective actions to maximize the reward.

One way to monitor the exploration–exploitation balance is by visualizing the policy entropy over time in a tensor board. An increasing trend in the entropy indicates that the policy is exploring more possibilities, whereas a decreasing trend indicates that the policy is becoming more deterministic and less exploratory. It was important to find the right balance between exploration and exploitation to achieve optimal performance in the environment, as can be seen in Figure 7. Policy entropy is calculated using the following formula:(8)H(p)=−∑(pi∗log(pi))
where pi is the probability of selecting action *i* according to the policy, and log(pi) is the natural logarithm of pi. The entropy is a measure of uncertainty associated with the policy’s action selection, and it ranges from 0 (when the policy always selects the same action) to log(N) (when the policy selects actions uniformly from a set of *N* possible actions).

### 5.2. Model Evaluation with Previous Studies

In the final step, the model was evaluated in comparison to previous studies. Although there are no similar purposes for using RL with AR, the studies mentioned in Table 2 used ML and RL for localization and tracking similar to our method. As stated before, Chan et al. [27] described a system for learning to paint in AR using GANs. The authors used the Adam optimizer with a learning rate of 0.0002 and a batch size of 32. They also used a cross-entropy loss function and reported a final accuracy of 87.2%.

In this context, the optimizer refers to the algorithm that is used to update the neural network weights during the training process. Xing et al. [23] also used the Adam optimizer with a learning rate set to 0.001 and a batch size of 32. This means that, in each iteration of the optimization algorithm, 32 training examples are used to update the neural network weights. Regarding accuracy, the authors evaluate the performance of their proposed method in terms of the average reward, which is a metric used for RL. They also compare their method with several baseline algorithms, including random offloading, heuristic-based methods, and deep learning-based methods.

In addition, Hetzel et al. [24] used the deep deterministic policy gradient (DDPG) algorithm with a replay buffer, the Adam optimizer with a learning rate of 0.001, and a batch size of 64 for their deep RL model. They evaluate the performance of their model by measuring the average reward obtained by the agent during training and testing and compare it with a baseline model that uses a fixed policy for typing on the virtual keyboard. The authors do not report a metric of accuracy for their deep RL model. Instead, they evaluate the performance of their model by measuring the average reward obtained by the agent during training and testing and compare it with a baseline model.

As can be seen in Table 3, the model’s advantages and strengths are reflected in the performance of the agent during the evaluation phase. It has been able to achieve the highest possible reward in each trial. The fact that the standard deviation is 0.000 indicates that there is no variability in the rewards obtained. In other words, every trial or episode in the task results in the same reward of 1.000. This situation is relatively rare in practice, as most tasks and environments involve some degree of variability in the rewards obtained. The model can effectively learn and adapt to the task of controlling the pipette device for use with the NanoDrop Spectrophotometer machine. The use of customizable hyperparameters, such as batch size, buffer size, learning rate, and number of epochs, allows for the dynamic control of the training process and optimization of the model’s performance. The incorporation of additional features such as normalization, layer depth, and encoding type, as well as reward signals and goal conditioning, suggests that the model is designed to effectively learn and adapt in complex environments. The application of this model to control a pipette device for use with a NanoDrop Spectrophotometer has potential real-world applications in the fields of biology, chemistry, and biotechnology.

In summary, our model appears to be a promising AR-assisted deep RL-based approach for controlling a pipette device for use with a NanoDrop Spectrophotometer. The high accuracy, customizable hyperparameters, and additional features make it suitable for a range of applications in scientific research and development.

### 5.3. Overall Performance

In RL, the environment is the external system with which the agent interacts. The environment provides feedback to the agent in the form of rewards and observations, and the agent’s goal is to learn a policy that maximizes the cumulative reward it receives over time. If the cumulative reward increases while the policy entropy also increases, it suggests that the model is making progress in learning to perform well in the environment while also maintaining good balance between exploration and exploitation. The increasing trend in the cumulative (Figure 8) reward indicates that the agent is learning and improving its behavior in the environment. This is a positive sign because it suggests that the agent is getting closer to achieving its goals and maximizing the reward. The increasing trend in policy entropy indicates that the agent is exploring more possibilities and trying out different actions to gain a better understanding of the environment. This is also a positive sign because it suggests that the agent is not getting stuck in a suboptimal solution and is continuing to learn and improve.

When the cumulative reward and policy entropy is increasing while the learning rate is also increasing, it suggests that the model is making progress in learning to perform well in the environment and is doing so at a faster rate. The fast learning rate indicates that the agent is being trained more aggressively, which can lead to faster convergence and better performance. The model is using a batch size of 10 and a buffer size of 100, which means that the agent is updating its policy every 10 steps and storing up to 100 past experiences for training. The learning rate is set to 0.0005 for the beta parameter and 0.0003 for the epsilon parameter, with a schedule that is linear for both parameters. The beta parameter controls the strength of the entropy regularization term, which encourages exploration during training, whereas the epsilon parameter controls the strength of the clip loss term, which prevents the policy from deviating too much from its previous policy.

## 6. Conclusions and Future Work

The paper proposed a system that uses the AR-assisted deep RL-based model for training and maintenance purposes that can be used in the I4.0 setting. The system uses an iPad camera to detect the target asset via feature matching, tracking techniques, and 3D modeling. Once the detection is complete, AR technologies generate clear and easily understandable instructions for the maintenance operator’s device. The development of the system followed virtualization, modularity, and service-oriented principles, creating a personalized digital training and maintenance service that can be adjusted to fit different manufacturing situations. The effectiveness in the application was evaluated on an actual NanoDrop Spectrophotometer in the industry, demonstrating that the use of 3D visualization for maintenance instructions enhances the operators’ user experience during maintenance tasks. The proposed model is well configured for the training using the PPO algorithm. By optimizing the learning rate, batch size, buffer size, and other hyperparameters, the model is designed to balance exploration and exploitation during training and converge to an optimal policy that maximizes the extrinsic reward.

The research result states that the model’s target technique with an AR-assisted deep RL-based approach resulted in a mean reward of 1.000 and a standard deviation of 0.000. This shows that all the rewards that were obtained in the given task or environment were exactly the same. The fact that the standard deviation is 0.000 indicates that there is no variability in the rewards obtained. Every trial in the task results in the same reward of 1.000. This situation is relatively rare in practice, as most tasks and environments involve some degree of variability in the rewards obtained. However, if this were the case, it would suggest that the agent (Pipette) is performing very well and achieving the maximum possible reward in every trial. This indicates that the model target with AR-assisted deep RL-based approach is a highly effective approach for achieving accurate results in the given task or application. This finding may be useful for researchers and practitioners that are interested in using AR-assisted deep RL-based model for machine learning or computer vision applications, as it suggests that the approach can lead to high levels of accuracy. However, it is important to note that the finding is limited by our specific task or studied application and may not necessarily generalize to other jobs or applications. Additionally, other factors, such as the quality of the data or the specific algorithm used, may have contributed to the observed accuracy. The finding provides valuable information about the effectiveness of the model target with the AR-assisted deep RL-based approach in the specific context of the study and may serve as a useful reference for future research and development in this area.

The future development of the application will concentrate on broadening its functionality, such as facilitating the integration of the application into a cloud-based server. This will allow operators to communicate with experts and access more intricate visualized instructions through their handheld devices. Moreover, the system is expected to offer managers the ability to devise personalized maintenance procedures. In addition to development, additional testing scenarios can be explored, including human participants, and evaluations with operators of varying levels of expertise. Several areas could be explored to further improve the model’s performance, such as improving the quality of the data, exploring different reward signals and conditioning methods, and experimenting with different architectures and network settings. By continuing to refine and optimize the model, it may be possible to achieve even better performance in the environment and advance the state of the art in RL.

## Figures and Tables

**Figure 1 sensors-23-06024-f001:**
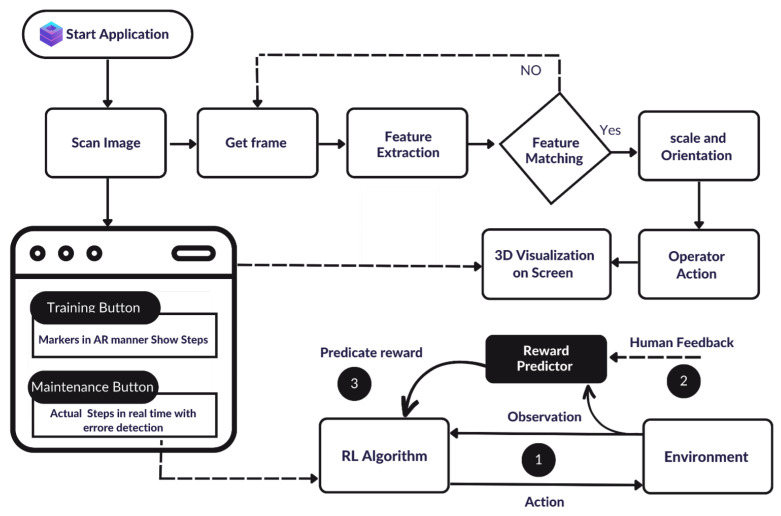
Flowchart of the proposed method for training and maintenance procedures using AR-assisted deep RL-based model that include: (1) scan image; (2) obtain frame; (3) feature extraction; (4) feature matching; (5) scaling and orientation; (6) operator action; (7) a 3D visualization on the screen; and (8) RL environment.

**Figure 2 sensors-23-06024-f002:**
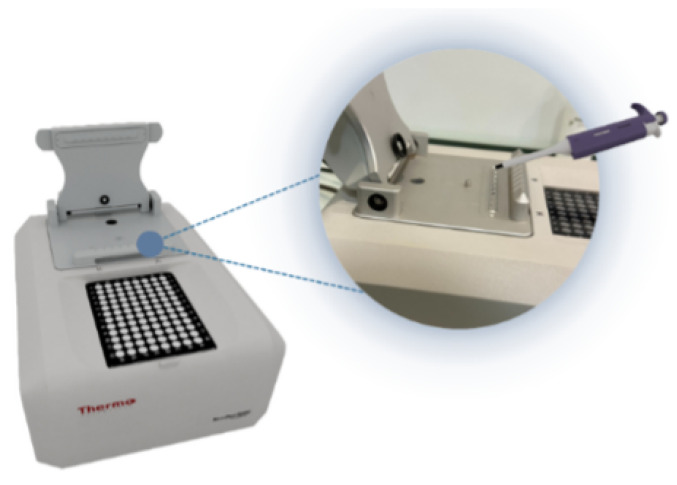
A 3D model representation of the NanoDrop Spectrophotometer device that is used as a case study for training and maintenance.

**Figure 3 sensors-23-06024-f003:**
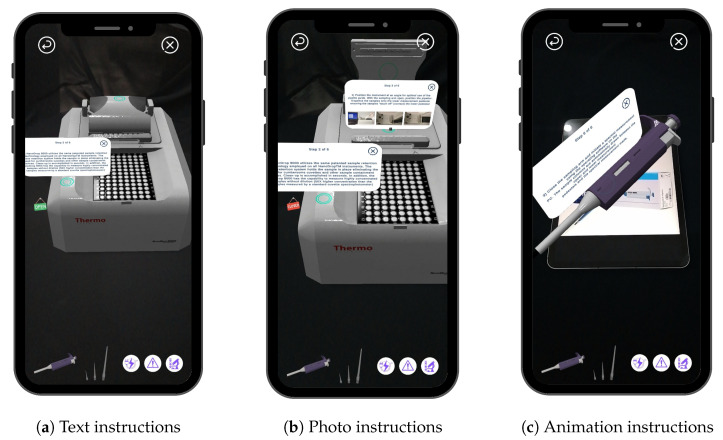
The application generated an exploded view of the NanoDrop Spectrophotometer, which highlights its subparts and aids in step-by-step training for operating the machine.

**Figure 4 sensors-23-06024-f004:**
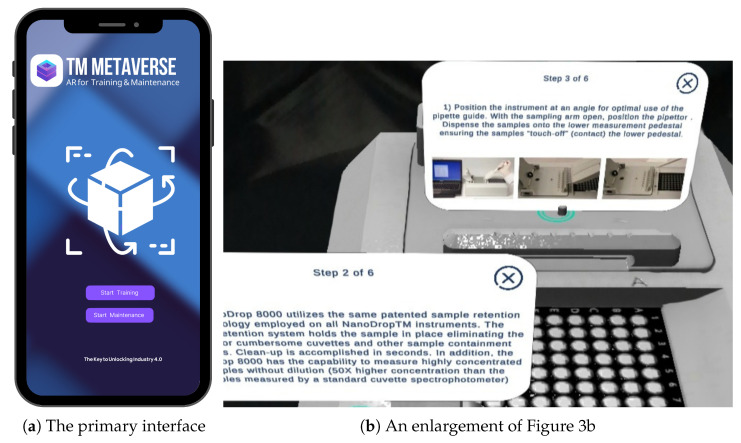
(**a**) The main interface with two button options for both scenarios: training and maintenance; and (**b**) The actual view seen by the user in the training scenario which is presented in Figure 3b.

**Figure 5 sensors-23-06024-f005:**
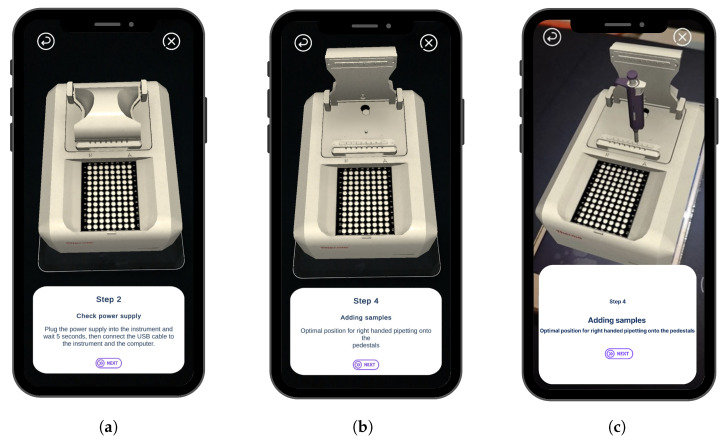
The steps involved in carrying out maintenance procedures with the NanoDrop Spectrophotometer using the application: (**a**) Showing text instructions for maintenance; (**b**) Showing the second step where the LED is open; and (**c**) Adjusting the angle of the pipette when adding samples.

**Figure 6 sensors-23-06024-f006:**
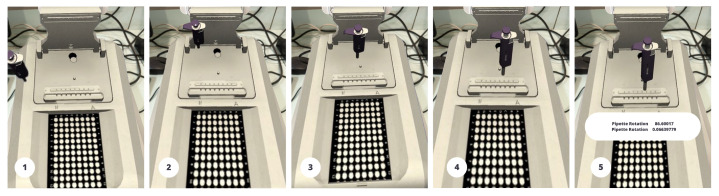
An animated movement of the pipette device as an RL agent on top of the NanoDrop Spectrophotometer as a target that keeps moving until the pipette reaches the correct position which helps maintain and adjust the pipette angle while adding the DNA samples.

**Figure 7 sensors-23-06024-f007:**
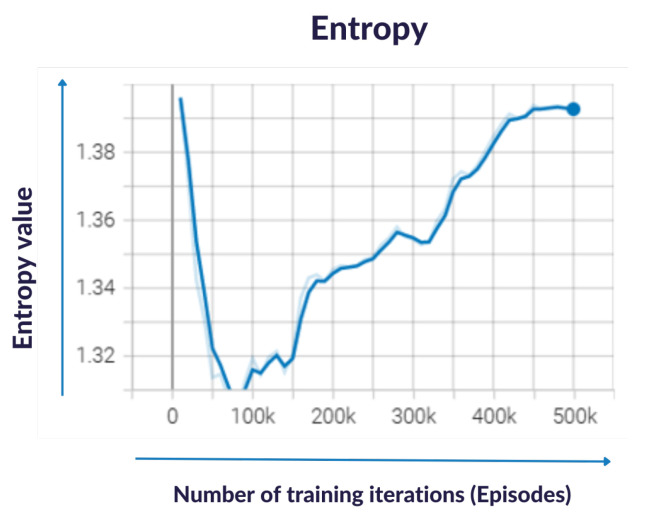
Result of exploitation of the process of choosing the most effective actions for the pipette while training with an increasing trend in entropy.

**Figure 8 sensors-23-06024-f008:**
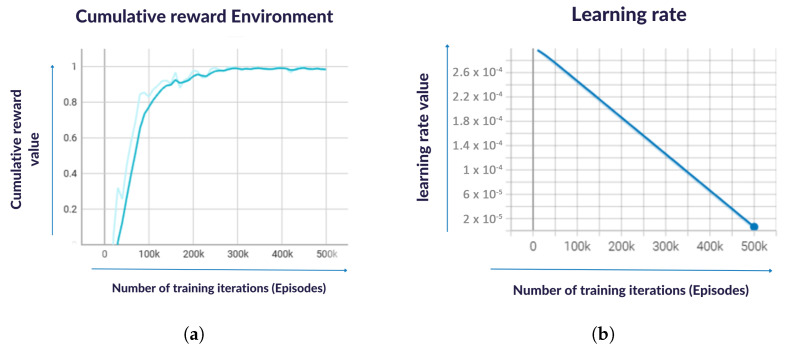
Result of the increasing trend in the (**a**) cumulative rewards over time and (**b**) learning rate on training progress of the AR-assisted RL based model.

**Table 1 sensors-23-06024-t001:** A summary of the outcomes of training and maintenance in previous studies including the I4.0 type, software, hardware, limitations, and challenges.

Ref	I4.0 Type	Outcome	Software	Hardware	Limitations and Challenges
Chengxi et al. [12]	AR framework	Deep RL framework that is powered by AR to provide a safe and reliable HCI	Unity 3D, MRTK Toolkit3 and Vuforia Plug-In	Hololens 2 AR headset and UR5 6-Joints robot with a gripper	Limited computing power and network condition of AR devices during scene information processing, policy control inference, control command synchronization
Chan et al. [27]	VR framework	Improve image quality and computational efficiency of 3D GANs	PyTorch and StyleGAN2	PC	Artifacts, lack finer details, and required knowledge of the camera pose
Xing et al. [23]	Markerless AR	Deep RL based on a multi-agent MADDPG to enable multi-user wireless network on an MEC server for AR application	PyTorch	Mobile devices and edge servers	The need for high-quality data to train the deep RL algorithm.
Hetzel et al. [24]	AR framework	First RL-based user model for mid-air and surface-aligned typing on a virtual keyboard	Python and various ML libraries	Leap motion controller and PC to run the RL algorithm	The reliance on a specific type of hand-tracking device and the collection of accurate and reliable hand-tracking data.
Malta et al. [18]	Automobile engines	Trained object model learns quickly and provides a prediction quickly	YOLOv5 deep neural network	AR Glasses	Trained model integration in the CMMS
Mourtzis et al. [31]	MR mobile application	Holographic images of the components via basic hand gestures	Cloud-based	Mobile, tablet, HMDs	The 3D component’s misalignment with respect to the actual machine
Mourtzis et al. [33]	MR controlling the robotic arm	Framework for controlling the robotic arm safely and remotely in almost real time	ROS	MR Microsoft HoloLens	Virtual machines and actual computers share resources, which compromises the framework’s speed
Alizadehsalehi et al. [15]	BIM to XR	Combines BIM and XRs to give access to all AEC stakeholders	Revit 2019/ cloud server	MR Microsoft HoloLens	The degree of stakeholder understanding of XR, software usability and willingness to pay extra for software, hardware, and training
Jang et al. [2]	MR touch hologram in midair	Using ultrasonic haptics with a fixed range	Ultimate immersion	VR HMD	Having the capacity to “feel” content in midair
Arena et al. [3]	AR-eye decides in medical applications	Makes the blood sampling easier by using portable scanners	Augmented reality markup language (ARML)/IOT	Wireless sensor network (WSN) and wireless body area network (WBAN)	Increases the AR system’s tracking precision
Ratcliffe et al. [34]	Remote XR	Restrictions in system development, data gathering, and participant recruiting	Unity 3D	AR-HMD (HoloLens)	Restrictions on data collection for remote XR
Surale et al. [32]	VR bare-hand midair mode-switching	The LEAP can accurately display hands and track their movements	Unity 3D	Standard HTC controller w	Incorrect classification of a dominant fist, palm, and pinch postures when rapidly switching mode; difficult to accurately recognize pinch actions that started manipulation

**Table 2 sensors-23-06024-t002:** Performance metrics for goal-oriented tasks: move-to-goal steps, time elapsed, mean reward, and standard deviation of rewards.

Move to Goal Steps	Time Elapsed (s)	Mean Reward	Std. of Rewards
Step: 10,000	55.449	−0.877	0.447
Step: 20,000	85.770	−0.093	0.787
Step: 30,000	117.238	0.323	0.857
Step: 40,000	148.027	0.691	0.599
Step: 50,000	179.527	0.578	0.645
Step: 60,000	210.429	0.704	0.539
Step: 70,000	241.473	0.707	0.552
Step: 80,000	272.897	0.680	0.578
Step: 90,000	304.707	0.708	0.569
Step: 100,000	336.163	0.863	0.432
Step: 200,000	656.214	0.983	0.129
Step: 300,000	982.363	0.989	0.146
Step: 400,000	1319.801	0.990	0.121
Step: 500,000	1671.575	1.000	0.000

**Table 3 sensors-23-06024-t003:** Model evaluation with previous work including the used approach, optimizer, learning rate, batch size, loss function, and accuracy.

Ref	Approach	Optimizer	Learning Rate	Batch Size	Loss Function	Accuracy
Chan et al. [27]	Geometry-aware 3D using GAN	Adam	0.0002	32	Cross-entropy	87.2%
Xing et al. [23]	Deep Q-learning approximates the Q-value function, which estimates the expected cumulative reward for taking a particular action in a particular state	Adam	0.001	32	Q-learning with mean squared error (MSE)	Average reward
Hetzel et al. [24]	RL with DDPG for virtual keyboard typing	Adam	0.001	64	DDPG	Average reward
Our Model	RL-PipTrack: AR-assisted deep RL-based model	PPO	0.0005	10	Clipped surrogate objective	Average reward + standard deviation

## Data Availability

Available on the first author’s GitHub at https://github.com/hq997/-AR-Assisted-Deep-RL-Based-Model (accessed on 31 May 2023).

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
