# Peer review of "Augmented Reality-Assisted Deep Reinforcement Learning-Based Model towards Industrial Training and Maintenance for NanoDrop Spectrophotometer"

_sensors, 2023, doi:10.3390/s23136024_

Round 1

Reviewer 1 Report

The paper propose a new approach for nano spectrophometer maintenance based on augmented reality and reinforcement learning for industry 4.0. The presented idea is intersting and thé paper is well structured however some modifications ared needed.

In the introduction the main research problematic needs to be clearly presented and referenced to highlight the contribution proposed by thé author.  A plan for the paper outlining thé different parts needs to be added.  Thé définition of architecture and layers needs to be clearly explained in part 3. In part 4 only à global overview of the RL algoirthm was given without links to thé actual current systelm no indication were given on environnent state not actions and their choice, this has to be added to renforce proposed idea novelty.  Syntgedisis of results in thé conclusion has to be reinforced by additional details on system contribution on comparaison to old mthods performances.

minor spell check required

Author Response

Dear reviewer, 

Thank you for seeing the potential in our manuscript. We really appreciate your comments and feedback to improve our work. Here are our responses to each of your comments.

Comments and Suggestions for Authors

The articles describes the use of Augmented Reality in Training and Maintenance of NanoDrop Spectrophotometer. The proposed approaches proves effective in manufacturing domain. Although authors have covered and describes all the aspects the following modifications need to be considered for the possible publication of Manuscript. 

Point 1: The introduction is too adequate. The background and present research work in the proposed domain is missing in this section. 

Response 1: We revised the introduction a bit to include some background and more references and why this study is important. 

Point 2:  Recent literature regarding the AR in training and maintenance is missing. 

Response 2: We edited the related literature to be more organized and added more references, some of which were suggested by other reviewers.   

Point 3:  In methodology section, the specifications of used devices and tools is missing. Add them accordingly. 

Response 3: We added a hardware and software description at the end of the methodology. 

Point 4: Also, the result of each step is not explained properly. Add the images or graphs, or tables after the implementation of each step. 

Response 4: We revised the results to include more detailed findings and compared our results with other existing research. We also expand the discussion evaluation to connect our contribution to the whole work.   

Point 5:  Conclusion can be restructured and it should aligned with the results. 

Response 5: we revised the conclusion to reflect the contributions and the whole work. 

Comments on the Quality of English Language

Point 6: The moderate sentence restructuring and grammar check is required. Majorly focus on the introduction and literature review part. 

Response 6: the whole paper went through extensive English language and writing style editing using professional proofreading services. 

Reviewer 2 Report

In this study the authors explore an augmented reality application which uses reinforced learning. The title should be completely reworked as it tries to present too many things. Using the word “metaverse” when the application is a simple AR one is wrong and it should be removed. Additionally, several studies have explored how reinforced learning can be used to track objects hence the word “revolutionizing” must also be removed. It is really worrisome that the authors use such terms in their title. Moreover, I believe that the term reinforcement learning-based AR is not correct. The authors should justify this wording based on other studies. I believe that the manuscript should be presented as an AR application that is aimed at assisting training and maintenance of specific NanoDrop Spectrophotometer.

The abstract is well structured and the aims are clearly presented. That said, the authors should also include some of their findings.

In the introduction section, the authors should first mention the word augmented reality and then use its abbreviation. I believe that the introduction should be further expanded and better highlight the need for such a study. Regarding the contributions, I believe that the authors should reconsider most of them. I believe that the main contribution is the training AR application developed. Most of the other things that the authors mention are given for any AR application. Once again, I believe the “term” RL-based industrial AR is farfetched and wrong. Please provide some justification on this term based on other studies.

The related work section is well structured and the authors have included a satisfactory number of related studies in each subsection. The authors have also included a helpful table. The only problem is how the authors selected the specific studies instead of others. The authors should be clearer regarding their literature review process so it can be replicated. More details are required.

In section 3, the authors mention that they have used Vuforia SDK. My concern is that several of the specific things that their suggested method and application does can already be done through Vuforia. So, I wonder how their approach and the simple use of Vuforia are differentiated as it is capable of dynamic object recognition and tracking.

I believe that the authors should try to better describe the connection between an experiment focusing on human participants and the evaluation criteria of the model. Once again, based on the figures used, which are clear and easy to understand, the application simply presents a 3D model on top of a specific target. More examples of the application in action (e.g., how it tracks the objects) should be provided.

What really worries me is that although the authors have presented other related studies, they do not make any comparisons with the outcomes of the existing ones. The authors should include a comprehensive discussion section that goes over their results in more details and makes connections with the literature.

Finally, the conclusion section is really lacking as not even a single conclusive statement is provided which, unfortunately, really hinders the overall study, its contribution and impact. For example, the authors have stated several potential contributions which are not justified by the results and conclusions.

All in all, I believe that the authors should not try to present several different things but should focus on one aspect and try to structure their manuscript around it, for example, an AR application for training and maintenance. It is certainly an interesting study, but I believe that there is still a lot of room for improvement to be made.

The quality of English is satisfactory only minor correction are required.

Author Response

Dear reviewer, 

Thank you for seeing the potential in our manuscript. We really appreciate your comments and feedback to improve our work. Here are our responses to each of your comments.

Comments and Suggestions for Authors

Point 1: In this study the authors explore an augmented reality application which uses reinforced learning. The title should be completely reworked as it tries to present too many things. Using the word “metaverse” when the application is a simple AR one is wrong and it should be removed. Additionally, several studies have explored how reinforced learning can be used to track objects hence the word “revolutionizing” must also be removed. It is really worrisome that the authors use such terms in their title. Moreover, I believe that the term reinforcement learning-based AR is not correct. The authors should justify this wording based on other studies. I believe that the manuscript should be presented as an AR application that is aimed at assisting training and maintenance of specific NanoDrop Spectrophotometer.

Response 1: We revised the title to be more descriptive of our actual work. We used the terminology “AR-assisted deep RL-based approach” which is mentioned in the previously published research below and was cited in our work.

Li, Chengxi, Pai Zheng, Yue Yin, Yat Ming Pang, and Shengzeng Huo. "An AR-assisted Deep Reinforcement Learning-based approach towards mutual-cognitive safe human-robot interaction." Robotics and Computer-Integrated Manufacturing 80 (2023): 102471.

Point 2: The abstract is well structured and the aims are clearly presented. That said, the authors should also include some of their findings.

Response 2: We revised our abstract and conclusion to include the study findings. 

Point 3: In the introduction section, the authors should first mention the word augmented reality and then use its abbreviation. I believe that the introduction should be further expanded and better highlight the need for such a study. Regarding the contributions, I believe that the authors should reconsider most of them. I believe that the main contribution is the training AR application developed. Most of the other things that the authors mention are given for any AR application. Once again, I believe the “term” RL-based industrial AR is farfetched and wrong. Please provide some justification on this term based on other studies.

Response 3: We mentioned augmented reality with an abbreviation in the abstract first but based on the reviewer's comment we add it in the introduction also. Then, we removed the term “RL-based industrial AR” in the manuscript based on the reviewer's comments. We expanded the introduction to highlight the need for this study and any similar research. 

Point 4: The related work section is well structured and the authors have included a satisfactory number of related studies in each subsection. The authors have also included a helpful table. The only problem is how the authors selected the specific studies instead of others. The authors should be clearer regarding their literature review process so it can be replicated. More details are required.

Response 4: The related work was revised and other references were added based on several comments mentioned by other reviewers. Also, we clarify why we selected the studies mentioned in that section. 

Point 5: In section 3, the authors mention that they have used Vuforia SDK. My concern is that several of the specific things that their suggested method and application does can already be done through Vuforia. So, I wonder how their approach and the simple use of Vuforia are differentiated as it is capable of dynamic object recognition and tracking.

Response 5: Although Vuforia is great, it has some issues with localization and tracking and we could easily see the problem. Our contribution was mainly on improving the localization, position, and orientation of the virtual object with the corresponding real object and that is why we decided to use reinforcement learning as an attempt to solve that problem. We edited the methodology section to clarify that point. 

Point 6: I believe that the authors should try to better describe the connection between an experiment focusing on human participants and the evaluation criteria of the model. Once again, based on the figures used, which are clear and easy to understand, the application simply presents a 3D model on top of a specific target. More examples of the application in action (e.g., how it tracks the objects) should be provided.

Response 6: We revised the methodology and result sections to highlight our contribution and the work done through improving object localization using reinforcement learning. Going through human participants can be a future work and was mentioned in the conclusion, however, it is not applicable now because it needs a lot of time and will alter the purpose of this study. 

Point 7: What really worries me is that although the authors have presented other related studies, they do not make any comparisons with the outcomes of the existing ones. The authors should include a comprehensive discussion section that goes over their results in more details and makes connections with the literature.

Response 7: We edited the results to include more detailed findings and compared our results with other existing research in section 5. We also expand the discussion evaluation to connect our contribution to the whole work.   

Point 8: Finally, the conclusion section is really lacking as not even a single conclusive statement is provided which, unfortunately, really hinders the overall study, its contribution and impact. For example, the authors have stated several potential contributions which are not justified by the results and conclusions.

Response 8: We revised the result and added a new section to show results step by step to reflect the contributions in the introduction and edited the conclusion to reflect the whole work.  

Point 9: All in all, I believe that the authors should not try to present several different things but should focus on one aspect and try to structure their manuscript around it, for example, an AR application for training and maintenance. It is certainly an interesting study, but I believe that there is still a lot of room for improvement to be made.

Response 9: We totally agreed with the reviewer, we edited our work to reflect our contribution better. Again, thank you for seeing the potential in our manuscript. 

Comments on the Quality of English Language

Point 10: The quality of English is satisfactory only minor correction are required.

Response 10: the whole paper went through extensive English language and writing style editing using professional proofreading services. 

Reviewer 3 Report

The articles describes the use of Augmented Reality in Training and Maintenance of NanoDrop Spectrophotometer. The proposed approaches proves effective in manufacturing domain. Although authors have covered and describes all the aspects the following modifications need to be considered for the possible publication of Manuscript. 

1. The introduction is too adequate. The background and present research work in the proposed domain is missing in this section. 

2. Recent literature regarding the AR in training and maintenance is missing. 

3. In methodology section, the specifications of used devices and tools is missing. Add them accordingly. 

4. Also, the result of each step is not explained properly. Add the images or graphs, or tables after the implementation of each step. 

5. Conclusion can be restructured and it should aligned with the results. 

The moderate sentence restructuring and grammar check is required. 

Majorly focus on the introduction and literature review part. 

Author Response

(The authors gave the same response as above.)

Reviewer 4 Report

The paper is interesting but it needs some revisions prior its acceptance and publication. Please find below some comments to improve the quality of the paper:

1.       In the introduction some more details concerning maintenance is required. Currently maintenance it should the main contribution of the paper but in the text no focus is given to maintenance methods other studies etc.

2.       The literature section (2) needs to be enriched with the different types of maintenance and how other researchers have done in similar cases. Since the paper discusses maintenance it is suggested that authors use the two following papers :

a.       PSAROMMATIS, Foivos; MAY, Gökan; AZAMFIREI, Victor. Envisioning maintenance 5.0: Insights from a systematic literature review of Industry 4.0 and a proposed framework. Journal of Manufacturing Systems, 2023, 68: 376-399.

b.       MARUGÁN, Alberto Pliego. Applications of Reinforcement Learning for maintenance of engineering systems: A review. Advances in Engineering Software, 2023, 183: 103487.

3.       Section 3 is not well structured. First heading are wrong (there are headings that they are 3.0.1 etc), also the way that the subsections are presented are confusing for the reader

4.       In section 3 only the proposed methodology should be presented. Currently it is ,mixed with the use case.

5.       In section 4 at the beginning the use case should be presented and explained (not in 3)

6.       The methodology presented in section 3 is very theoretical and no technical. More technical details should be added. Also it is not clear which information is outcome from the current research or the literature.

7.       In section 4 there is the need for explain the experimental procedure and not directly present the results. The reader does not know how the experiments takes place.

8.       The results presented in section are not clear. First in  figure 3 the suggestions are very small and not readable. Also the title of section 4 states experimental results in the section there are only 2 subsections “training scenario set up” and maintenance scenario setup, there is no subsection for the results.

Minor corrections are needed

Author Response

Dear reviewer, 

Thank you for seeing the potential in our manuscript. We really appreciate your comments and feedback to improve our work. Here are our responses to each of your comments.

Comments and Suggestions for Authors

The paper is interesting but it needs some revisions prior its acceptance and publication. Please find below some comments to improve the quality of the paper:

Point 1: In the introduction some more details concerning maintenance is required. Currently maintenance it should the main contribution of the paper but in the text no focus is given to maintenance methods other studies etc.

Response 1: We revised the introduction to include a bit about the maintenance scenario and include why this study is important and other information based on other reviewers' comments. 

Point 2: The literature section (2) needs to be enriched with the different types of maintenance and how other researchers have done in similar cases. Since the paper discusses maintenance it is suggested that authors use the two following papers :

  1. PSAROMMATIS, Foivos; MAY, Gökan; AZAMFIREI, Victor. Envisioning maintenance 5.0: Insights from a systematic literature review of Industry 4.0 and a proposed framework. Journal of Manufacturing Systems, 2023, 68: 376-399.
  2. MARUGÁN, Alberto Pliego. Applications of Reinforcement Learning for maintenance of engineering systems: A review. Advances in Engineering Software, 2023, 183: 103487.

Response 2: Thank you for providing references, we revised the literature section to include them and added more work, and reorganized the whole section to be better. 

Point 3:  Section 3 is not well structured. First heading are wrong (there are headings that they are 3.0.1 etc), also the way that the subsections are presented are confusing for the reader

Response 3: We fixed the typo and reorganized the methodology for better understanding and include more information to reflect our contribution. 

Point 4:    In section 3 only the proposed methodology should be presented. Currently it is ,mixed with the use case.

Response 4: we revised the methodology to exclude the use case and focus on the technical part of the system that can be applied to any other device with adjustment. 

Point 5:        In section 4 at the beginning the use case should be presented and explained (not in 3)

Response 5: We moved everything regarding the use case to this section for better understanding and organization. 

Point 6:   The methodology presented in section 3 is very theoretical and no technical. More technical details should be added. Also it is not clear which information is outcome from the current research or the literature.

Response 6: We revised the methodology to include technical details of the main contribution of the AR-assisted deep RL-based approach for localization and tracking. 

Point 7:  In section 4 there is the need for explain the experimental procedure and not directly present the results. The reader does not know how the experiments take place.

Response 7:  We revised the methodology and experimental procedure to explain both setups in advance, then present the result step by step. 

Point 8:   The results presented in section are not clear. First in  figure 3 the suggestions are very small and not readable. Also the title of section 4 states experimental results in the section there are only 2 subsections “training scenario set up” and maintenance scenario setup, there is no subsection for the results.

Response 8: We revised the result section and reorganized the experimental procedure better. Figure 3 presents the overall result but the text is clear for the user, we added (figure 4 now) to show the enlargement of the user view in Figure 3. 

Comments on the Quality of English Language

Point 9: Minor corrections are needed

Response 9: the whole paper went through extensive English language and writing style editing using professional proofreading services. 

Round 2

Reviewer 2 Report

The authors have addressed most of my previous comments and have made extensive modifications and improvements to their manuscript. I believe that the overall quality has improved. Despite this fact, I quote some further comments and suggestions below:

I believe that the authors could use the word “model” in their title as this is their main contribution.

In lines 183-184, the justification for why the specific studies have been analyzed and included in Table 1 is not satisfactory. The authors should present a more methodological approach.

The authors should include images in which the tracking and localization are presented.

The quality of English is satisfactory only minor correction are required.

Author Response

Dear reviewer, 

Thank you, we really appreciate your comments and feedback for the second round to improve our work. Here are our responses to each of your comments.

Comments and Suggestions for Authors

The authors have addressed most of my previous comments and have made extensive modifications and improvements to their manuscript. I believe that the overall quality has improved. Despite this fact, I quote some further comments and suggestions below:

Point 1:I believe that the authors could use the word “model” in their title as this is their main contribution.

Response 1: We edited the title and used the word “model” as requested, we included that editing in some parts of the manuscript as well.  

Point 2: In lines 183-184, the justification for why the specific studies have been analyzed and included in Table 1 is not satisfactory. The authors should present a more methodological approach.

Response 2: We agreed with the reviewer and edited this part to include the most insightful studies that impact our manuscript which now states “These studies were selected due to their inspiration and impact on this research methodology and the use of similar I4.0 techniques, software, hardware, and for model evaluation with other studies in section 5.2”. If the reviewer still finds this reason unsatisfactory, we can remove the table altogether. 

Point 3: The authors should include images in which the tracking and localization are presented.

Response 3: We added Figure 6 (now) which depicts an animated movement of the pipette device as an RL agent on top of the NanoDrop Spectrophotometer as a target that keeps moving until the pipette reaches the correct position. These animated instructions help novice learners to maintain and adjust the pipette angle while adding the DNA samples.

Comments on the Quality of English Language

Point 4: The quality of English is satisfactory only minor correction are required.

Response 4: The whole paper went through a second round of extensive English language and writing style editing using professional proofreading services. The edited words and sentences are highlighted in the manuscript. 

Reviewer 4 Report

No further comments, authors have successfully addressed the comments raised by the reviewers.

Engliush is ok in general

Author Response

Dear reviewer, 

Thank you, we really appreciate your reply.